# Preliminary Investigation about *Aspergillus* spp. Spread in Umbrian Avian Farms

**DOI:** 10.3390/jof8111213

**Published:** 2022-11-16

**Authors:** Deborah Cruciani, Silvia Crotti, Carmen Maresca, Ivan Pecorelli, Emanuela Verdini, Marinella Rodolfi, Eleonora Scoccia, Sara Spina, Andrea Valentini, Francesco Agnetti

**Affiliations:** 1Istituto Zooprofilattico Sperimentale dell’Umbria e delle Marche “Togo Rosati” (IZSUM), 06126 Perugia, Italy; 2Department of Earth and Environmental Sciences, University of Pavia (UNIPV), 27100 Pavia, Italy

**Keywords:** aflatoxin B_1_, *Aspergillus* spp., birds, farm, poultry

## Abstract

Among the fungi responsible for deep mycosis, the genus *Aspergillus* plays a predominant role both in human and veterinary medicine. From a “One Health” perspective, infections by *Aspergillus* spp. often represent a public health problem linked to specific occupational categories that could have a greater risk of inhaling spores and developing any respiratory disease. This preliminary investigation allowed to acquire information about the spread of *Aspergillus* spp. in avian livestock of the Umbria region (Central Italy), their sensitivity to antifungals, and the presence of mycotoxins in the considered farms. Environmental, feed, animal, and human samples were collected for mycological investigations; chemical analyses were also performed in feed samples. Moreover, prevalence estimated of the fungal isolates were provided for each individual farm sampled. Direct fungal identification was possible in 298 out of the 559 total samples; 162 of the samples were positive for *Aspergillus* spp. Mycotoxins were detected in 5 out of the 21 feed samples collected. All the aspergilli tested for antifungal susceptibility were resistant to fluconazole. The results obtained show how much the genus *Aspergillus* is widespread in the investigated farms; therefore, the poultry livestock represents a favorable environment for the maintenance and spread of fungal spores and their potential transmission to animals and humans.

## 1. Introduction

In the last decades, mycoses have acquired a considerable clinical importance. Nowadays, human mortality due to invasive fungal infections is comparable to that induced by tuberculosis or malaria worldwide [1,2,3]. The global increase in fungal diseases is mainly related to the better ability to diagnose them, the greater number of immunocompromised patients, the increasingly frequent use of invasive medical procedures, as well as prolonged antibiotic or corticosteroid therapies. Concerning animals, different groups such as pet, livestock, and wild species are widely affected by fungal agents too.

Deep and systemic mycoses play a more worrying role in public health, given their incidence, their clinical evolution, and their often difficult therapeutic management [4]. Furthermore, many fungal organisms often implicated in deep mycosis are intrinsically resistant to common antifungals [5].

The genus *Aspergillus* plays a preponderant role among the fungi responsible for deep mycosis; in particular, *A. fumigatus* can be responsible of bronchopulmonary allergic forms, especially in humans, worsening the clinical signs in patients with asthma, cystic fibrosis, or Chronic Pulmonary Aspergillosis (CPA). Allergic Bronchopulmonary Aspergillosis (ABPA) is still underestimated from a diagnostic point of view; its clinical or radiological features are confused with pulmonary tuberculosis, especially in developing countries. In industrialized countries, the estimated incidence of the disease is very high (e.g., in Europe about one million patients). From a pathogenesis perspective, recurrent exposure to high concentrations of *A. fumigatus* spores is the most cited risk factor in the development of ABPA, even if not the only one [6,7]. Also in veterinary medicine, *A. fumigatus* is associated with deep fungal respiratory infections, almost exclusively in birds [8]: exposure to high concentrations of environmental spores is the determining predisposing factor for the onset of the disease, even in this case. Hot-humid farm environments, often overcrowded and lacking in ventilation and hygiene, could contribute to the proliferation of these fungi and increase the exposure of animals to the infection, as well as poorly stored foodstuffs. The immune status of birds, prolonged therapies with antibiotics or corticosteroids, vaccinations, and metabolic disorders also represent further predisposing factors [9]. *Aspergillus* spp. is also mentioned in cases of dog rhinitis, mycosis of the guttural pockets in horses, bovine abortions and pneumonia in dolphins; however, these clinical conditions are uncommon [10,11]. The greater susceptibility of birds compared with mammals could be explained by differences between the innate and the acquired immunity against fungi, as well as the anatomical features of birds, such as a lack of epiglottis and a diaphragm or limited distribution of pseudo stratified columnar ciliated cells in the respiratory tract which could facilitate fungal engraftment [8]. Therefore, from a “One Health” perspective, deep infections by *Aspergillus* spp. can have an impact to public health. In particular, they represent a problem linked to specific occupational fields because some workers (e.g., veterinarians, breeders, poultry farms or feed mills operators) could have a greater risk of inhaling spores and developing any respiratory conditions [12].

Particularly, thanks to a multidisciplinary approach, this investigation provided information relating to: (i) *Aspergillus* spp. spread in some avian farms of the Umbria region (Central Italy); (ii) their sensitivity to common antifungals; (iii) the presence and spread of mycotoxins in feed samples. The data obtained have been useful to understand the potential risk of spreading and transmitting fungal spores and air-borne mycotoxins to workers operating in the poultry sector.

## 2. Materials and Methods

Through the involvement of the territorial Veterinary Services, eight poultry farms located in the Umbria region were selected for the investigations as follows: four intensive farms in the province of Perugia, three intensive farms, and one rural farm in the province of Terni. All farms were constantly monitored for the main infectious diseases, such as Influenza, West Nile Disease, and salmonellosis.

In the intensive farms, broilers (*Gallus gallus*) were raised indoors, in sheds ranging from 400 to 800 m^2^ (Appendix A), with controlled ventilation and temperature; animal density was about 20 subjects per m^2^; the litter was changed every two months, at the end of the production cycle; the feed was commercial and stored in special silos adjacent to the sheds; two operators worked six days a week in each farm.

In the rural farm, broilers, guinea fowl (*Numida meleagris*), and ducks (*Anas platyrhynchos*) were bred in stables with a surface of about 40 m^2^ and the possibility of access to outdoor parks (Appendix A); there was not monitoring of the ventilation or temperature inside the stables; the litter was changed approximately every two months; the feed was self-produced and stored in a closed room adjacent to the stables; there was only one operator responsible for the farm.

### 2.1. Sampling

For the isolation and the identification of *Aspergillus* spp. strains, the following specimens were collected: environmental samples, represented by air; surface swabs (from the walls of the sheds), litter, and soil; feed samples, directly collected from the silos and the feeders inside the sheds; animal samples, represented by respiratory mucosal swabs and pools of trachea, lungs, and air bags of birds found dead in the investigated farms; and human samples, represented by nasal swabs from farm workers.

The collection was planned on a monthly basis, in two periods of the year, one colder and one warmer: January–April and June–September. All the samples were stored in sterile containers until their transfer to the laboratory.

For mycotoxins analysis, to ensure representativeness of the entire feed lot, sampling was conducted according to EU Commission Regulation (EC) n. 152/2009.

### 2.2. Isolation of Aspergillus spp.

#### 2.2.1. Environmental Samples: Air and Surface Swabs

A quantitative mycological analysis was carried out at the IZSUM for the air monitoring and for the surface swabs. Czapek agar (CZA) stimulating the growth of xerophilic species deeply colonizers of substrates was used as a selective reference medium for *Aspergillus* spp. and *Penicillium* spp. For air and swabs samples, the gravity sedimentation method settling plates exposed for 20 min, and plating by dilution were the techniques used. For air samples, plates were incubated at 25 ± 1 °C in aerobiosis, whereas for swabs the incubation was performed at 37 ± 1 °C in aerobiosis, according to the laboratory standards provided for fungi. The observation of the colonies was performed within 72–96 h.

#### 2.2.2. Environmental Samples: Litter, Soil, and Feed Samples

A qualitative mycological examination was performed at UNIPV for litter, soil, and feed samples. The method used was direct-plating onto CZA. Plates were incubated at 25 ± 1 °C in aerobiosis, prolonging the incubation until 14 days in order to detect the slower growing strains too, and Petri plates replicates were constantly observed at low magnification (4×).

#### 2.2.3. Animal and Human Samples

A qualitative mycological examination was carried out at IZSUM from collected animal organs or human nasal swabs, as previously described. Samples were direct-plated onto CZA and incubated at 25 ± 1 °C and 37 ± 1 °C in aerobiosis. Colonies were daily observed within 72–96 h.

### 2.3. Identification of Aspergillus spp. and Evaluation of the Antifungal Resistance

In accordance with the specific monographs, the morphological identification of the isolated aspergilli was carried out through the evaluation of their macroscopic and microscopic features such as growth rate, shape, texture, colour, pigments or exudates of the colonies, micro-morphology of the conidiophores and the conidia, and any other fruiting organs and cellular organelles.

They were morphologically analysed according to Klich M. A. [13], and also referring to the most recent classification of Houbraken J. et al. [14].

A total of 12 strains were subjected to DNA extraction for molecular typing using QIAamp DNA mini kit (QIAGEN^®^, Valencia, CA, USA) following the initial analysis step with Lysozyme Buffer (Appendix A). Amplification of the β-tubulin gene, specific for non-dermatophytic molds belonging to the genus *Aspergillus*, was subsequently carried out [15]. The amplification product was then sequenced through the automated Sanger method to reach species identification. To further confirm the results, a Neighbor-Joining phylogenetic tree has been constructed using the Maximum Composite Likelihood method (MEGA11 Sofware). These strains were also tested for the sensitivity to the most common antifungals with semi-automatic method of micro-dilution in broth using Sensititre^TM^ YeastOne YO10 (Thermo Fisher Scientific, Waltham, MA, USA): the evaluation of the M.I.C. (Minimal Inhibitory Concentration) ensure compliance with CLSI recommendation [16,17]. The antifungal molecules tested were: anidulafungin, micafungin, caspofungin, 5-fluorocytosine, posaconazole, voriconazole, itraconazole, fluconazole, and amphotericin B.

### 2.4. Evaluation of the Presence of Mycotoxins in the Feed

The presence of mycotoxins in 21 poultry feed samples collected from the investigated farms was evaluated by IZSUM using an ISO 17025 [18] accredited method for the simultaneous determination of Aflatoxin B_1_, Ochratoxin A, and Zearalenone in feeds for zootechnical use and raw materials. Limits of quantification (LOQs) were set at 0.0020, 0.025, and 0.050 mg/kg for Aflatoxin B_1_, Ochratoxin A, and Zearalenone, respectively. After shredding in slurry form, the samples were extracted by MeOH/H_2_O (80/20 *v*:*v*) and purified with Immunoaffinity columns (IAC), which ensured maximum specificity and allowed concentrations to be reached well below the levels set by current EU regulations: Directive 2002/32/EC of the European Parliament and Commission Recommendation (EU) 2016/1319. The instrumental determination was carried out by Ultra High Performance Liquid Chromatography couplet with fluorescence detector (UHPLC-FLUO).

### 2.5. Statistical Data Processing

A descriptive analysis of the obtained information was carried out considering the farms and the types of samples. In addition, the prevalence and the relative 95% confidence interval (95% CI) for each tested farm were calculated.

The antifungal susceptibility of the isolated aspergilli from the animals and the environment as well as the possible presence of mycotoxins in the feed samples examined were evaluated (in particular *A. fumigatus* ones).

## 3. Results

### 3.1. Isolation and Identification of Aspergillus spp. and Evaluation of The Antifungal Resistance

In the eight selected farms, a total of 559 samples were collected and represented as follows: 394 environmental samples (121 air, 124 surface swabs, 112 litter, and 37 soil), 38 animal samples, 46 feed samples, and 81 human samples (Table 1).

Fungal growth was recorded in 298 out of the 559 total samples (53%, Table 2). As specifically regards the air and the surface swabs samples, the mycological positivity was expressed as colony-forming units, according to the current legislation [19,20]. Table 2 also shows a total of 162 strains (29%) with macroscopic and microscopic features referable to *Aspergillus* spp.: in particular, 26 (5%), 5 (0.9%), and 3 (0.5%) of them were identified as members of *A. fumigatus* complex, *A. flavus* complex, and black aspergilli (*A. niger* complex), respectively.

Macroscopically, colonies referable to *A. fumigatus* complex grown velvety onto CZA appeared green to blue to greyish in colour and edged in white, with a cream-colored reverse (Figure 1a,b). Microscopically, the characteristics observed perfectly agreed with those reported by monographs: hyphae hyaline and septate, conidiophores with smooth wall, vesicles dome or claviform with phialides originating directly on the upper half or 2/3 of the vesicle, conidia with small spikes covering their surface and produced in basipetal column chains (Figure 1c).

*A. flavus* complex colonies appeared yellow to yellow-green in colour, with a cream-colored reverse; the surface was grainy, often with radial grooves and edged in white (Figure 2a,b). Microscopically, hyphae hyaline and septate, conidiophores with rough wall, vesicles dome or claviform with phialides originating, both directly and from *metulae*; on the entire vesicle surface, spherical and echinulated conidia were noticed (Figure 2c).

Young sterile colonies referable to black aspergilli appeared white-yellow in colour at maturity, covered by a layer of black conidial heads with a cream-colored reverse; the surface was grainy and edged in white (Figure 3a,b). Microscopically, hyphae hyaline and septate, smooth conidiophores with a tick wall, spherical vesicles with phialides originating from *metulae* on the entire vesicle surface, spherical/oval rough conidia were observed (Figure 3c).

In Table 3 the distribution of positive samples for *Aspergillus* spp. in the investigated farms is showed.

The three aspergilli species isolated through mycological examination were confirmed by PCR and Sanger sequencing. In particular, the molecular investigation was performed on 4 *A. fumigatus* complex strains (one for each sample category), 5 *A. flavus* complex strains, and 3 *A. niger* complex strains. The consensus sequences obtained showed significant similarity with those registered in the Westerdijk Fungal Biodiversity Institute database for *A. fumigatus*, *A. flavus*, and *A. niger* (Table 4); these results were also supported by the phylogenetic analysis (Figure 4).

The results of the antifungal susceptibility testing are shown in Table 5: *Aspergillus* spp. strains showed resistance to all the antimycotic drugs, particularly to fluconazole (*n* = 12, 100%), 5-fluorocytosine (*n* = 8, 67%), and anidulafungin, micafungin, and caspofungin (*n* = 6, 50%).

### 3.2. Evaluation of the Presence of Mycotoxins in Feed

The investigation on mycotoxins (Aflatoxin B_1_, Ochratoxin A and Zearalenone) was carried out on 21 out of 46 feed samples, and 5 of them, all collected at the rural farm using self-produced feed, presented mycotoxins at concentrations above LOQs. In four of them Aflatoxin B_1_ was detected while in the other one both Aflatoxin B_1_ and Zearalenone were found. Although, the concentrations of the mycotoxins (0.0040 mg/kg, 0.0103 mg/kg, 0.0028 mg/kg and 0.0091 mg/kg for Aflatoxin B_1_; 0.054 mg/kg for Zearalenone) were below the limits established by current EU legislation the presence of these molecules (100% of the samples collected at the rural farm) is of high concern.

## 4. Discussion

This preliminary investigation aimed to evaluate an often underestimated public health problem through a multidisciplinary approach and different technical skills. Nowadays, deep or systemic mycoses play a more worrying role, both in the human and veterinary medicine, given their incidence, their clinical evolution, and their often-difficult therapeutic management. Moreover, fungal infections can be difficult to overcome due to few effective molecules available and the possibility that resistance phenomena arise. Especially in human medicine they can be involved in occupational diseases [21].

Despite the small number of poultry farms participating in the study, some considerations can be made.

Fungal species are widespread in the avian livestock investigated (53%). In particular, the genus *Aspergillus* was isolated in 162 of the total samples analysed (29%), with a high percentage if compared with the total of fungi observed (54%). This evidence agree with data emerged from other studies carried out in some European countries [12,22]. Combining conventional and molecular diagnostic approaches, 34 strains (21%) were identified as members of three of the major complexes threatening human and animal health. In detail, 4 *A. fumigatus*, 5 *A. flavus*, and 3 *A. niger* has been sequenced from environmental, feed, animal, and human samples with significant similarity to the sequences deposited in the specific database. The importance of these three species was given according to the literature data about respiratory fungal diseases in humans and animals [6,9]. The molecular analyses proposed in this investigation helped to overcome the limitation of conventional methods in the fungal discrimination compared with the most recent introduced taxonomy [23].

In seven investigated farms the prevalence of aspergilli ranging from 19% to 43%; only in the remaining one no aspergilli were isolated, even if this intensive farm was the less sampled (14 samples out of 559, 2.5%). The only rural farm showed a prevalence of 28%, with 20 positive samples for *Aspergillus* spp. in a total of 72 collected ones. Despite the small amount of the investigated farms, the high prevalence of fungal growth, approximately 50% in the “C” intensive farm, encourages similar studies.

In the present survey, the rural poultry farm used a self-produced feed, stored on the floor of a room adjacent to the stables where the poultry were raised, whereas in the intensive farms the commercial feed was stored in silos. Feed storage in an environment without monitoring in temperature, humidity, ventilation, or the presence of fungal spores can easily allow its contamination by molds, including mycotoxin-producing *Aspergilli*. Storage in silos can prevent the mycotoxins production as long as biosafety guidelines will be respected. Mycotoxins can spread as free air-borne particles or through spores as vectors, thus becoming inhalable for animals and workers with serious health implications [24]. The mycotoxins values found in this survey fell within the limits established by current European legislation. Nevertheless, they must be considered as “indicators” of the degree of farm environmental mycotic contamination.

In a total of 81 human nasal swabs, 20 (25%) showed fungal growth, 18/20 (90%) has been identified as *Aspergillus* spp. and 6/18 (33%) as *A. fumigatus* (*n* = 4, 67%) and *A. flavus* (*n* = 2, 33%). These fungi were found in the upper respiratory tract, but no cases of deep mycoses were reported. Despite this, all the strains subjected to antifungal susceptibility showed high percentages of resistance to common antifungals, especially fluconazole (100%) [25,26]. Periodical mycological investigations could represent a good tool to monitor the degree of the environmental contamination by fungal spores and enhance any corrective or preventive measures to limit ABPA forms or other respiratory mycotic diseases.

Finally, further studies are necessary including more farms with different management (intensive and rural) to improve the monitoring of fungi and correlate the isolated strains with those of human origin, especially considering the resistance to common antifungal molecules.

Many *Aspergillus* species (particularly belonging to sections *Nigri* and *Flavi*) are also highly used for enzyme production, food fermentations, biotechnology, and production of pharmaceuticals [27]. In industry and laboratory contexts, they are clearly harmless, and their contribution is advantageous considering that their use complies with the safety regulations and implies the use of individual and collective protection devices. To support this, data obtained in this preliminary investigation highlight the importance of biosafety also in management farm. Some precautions can be adopted to reduce the fungal spread, especially aspergilli, such as i) a correct feed storage to avoid mycotoxins production, (ii) an appropriate animal density to avoid overcrowding and to reduce the circulation of infective agents; (iii) the use of individual protection devices (filter masks, visors, disposable gloves and coveralls, etc.) to protect the farm workers; (iv) the introduction of periodic monitoring of the operators for a self-control plan.

## Figures and Tables

**Figure 1 jof-08-01213-f001:**
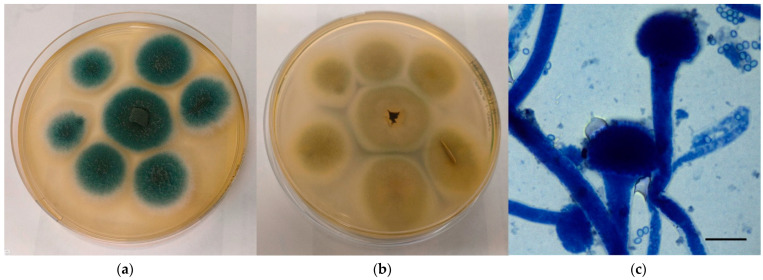
Mycological examination of the colonies referable to *A. fumigatus* complex grown onto CZA. (**a**) Macroscopic surface of the colonies; (**b**) Macroscopic reverse of the colonies; (**c**) Microscopic features of the colonies (methylene blue staining, ×100, bar 10 µm).

**Figure 2 jof-08-01213-f002:**
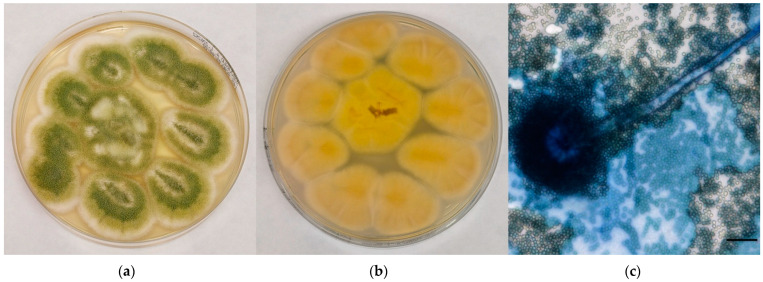
Mycological examination of the colonies referable to *A. flavus* complex grown onto CZA. (**a**) Macroscopic surface of the colonies; (**b**) Macroscopic reverse of the colonies; (**c**) Microscopic features of the colonies (methylene blue staining, ×40, bar 20 µm).

**Figure 3 jof-08-01213-f003:**
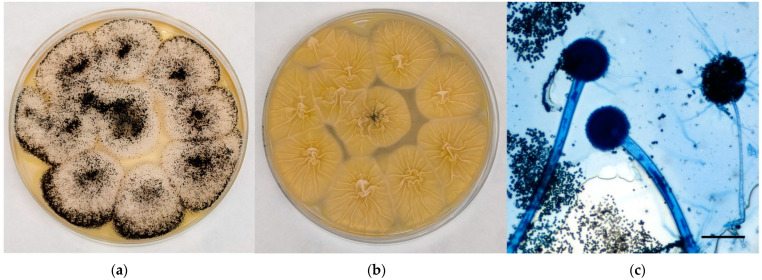
Mycological examination of the colonies referable to black aspergilli grown onto CZA. (**a**) Macroscopic surface of the colonies; (**b**) Macroscopic reverse of the colonies; (**c**) Microscopic features of the colonies (methylene blue staining, ×20, bar 50 µm).

**Figure 4 jof-08-01213-f004:**
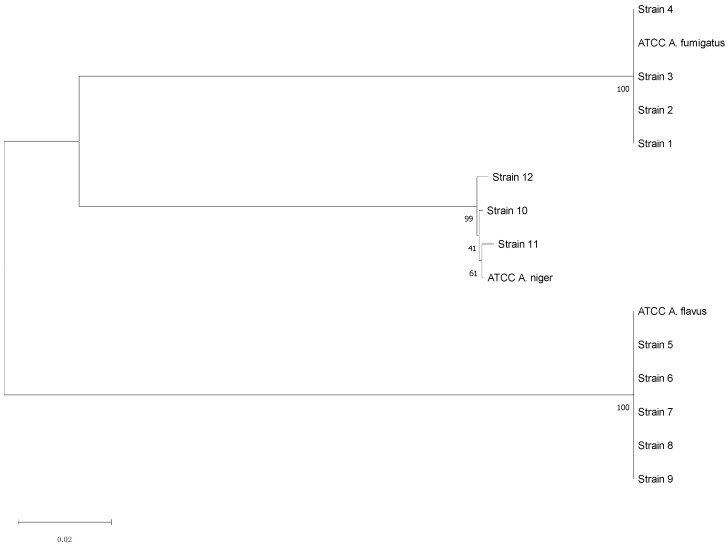
Neighbor-Joining phylogenetic tree of the sequenced *Aspergillus* strains. Maximum Composite Likelihood method, 1000 bootstraps (MEGA11 Software).

**Table 1 jof-08-01213-t001:** Sample distribution by farm. Legend: i = intensive; r = rural.

Farm/Tipology	Environmental Samples				Feed Samples	Animal Samples	Human Samples	Total Samples
	Air Samples	Surface Samples	Litter Samples	Soil Samples				
A/i	3	3	3	1	1	1	2	14
B/i	12	12	12	4	4	4	10	58
C/i	6	6	6	2	2	2	6	30
D/i	19	22	10	3	5	3	7	69
E/r	15	15	15	5	7	5	10	72
F/i	12	12	12	4	9	5	8	62
G/i	27	27	27	9	9	9	20	128
H/i	27	27	27	9	9	9	18	126
**Total samples**	**121**	**124**	**112**	**37**	**46**	**38**	**81**	**559**
	**394**							

**Table 2 jof-08-01213-t002:** Results of mycological examination.

Sample Category	Fungal Growth	*Aspergillus* spp.	*A.* *fumigatus* Complex	*A.* *flavus* Complex	*A.* *niger* Complex	No Fungal Growth	Total Samples
Environmental samples	238	111	16	2	2	156	394
Feed samples	34	30	4	0	1	12	46
Animal samples	6	3	2	1	0	32	38
Human samples	20	18	4	2	0	61	81
**Total *Aspergillus* spp.**		**162 (29%)**	**26 (5%)**	**5 (0.9%)**	**3 (0.5%)**		
**Total samples**	**298 (53%)**					**261 (47%)**	**559**

**Table 3 jof-08-01213-t003:** *Aspergillus* spp. prevalence distribution by farm. Legend: i = intensive; r = rural.

Farm/Tipology	*Aspergillus* spp. Positive Samples	Total Samples/Farm	Prevalence	95% CI
A/i	-	14	0%	-
B/i	11	58	19%	10–31%
C/i	13	30	43%	25–63%
D/i	26	69	38%	26–50%
E/r	20	72	28%	18–40%
F/i	19	62	31%	20–44%
G/i	46	128	36%	28–45%
H/i	27	126	21%	15–30%
**Total samples**	**162**	**559**	**29%**	**26**–**33%**

**Table 4 jof-08-01213-t004:** Results of aspergilli sequencing.

*Aspergillus* Species	Strain	Sample Category	Sample Type	Accession Number	Overlap	Similarity	Probability
*A. fumigatus*	1	Environment	Air	CNRMA13.349	98.07%	100.00%	0
	2	Feed	Feed	CNRMA20.388	98.05%	100.00%	0
	3	Animals	Lung	CNRMA13.349	98.25%	100.00%	0
	4	Humans	Nasal swab	CNRMA20.141	99.37%	100.00%	0
*A. flavus*	5	Environment	Air	UOA/HCPF 8374A	98.25%	100.00%	0
	6	Environment	Air	CBS 120.62	98.02%	100.00%	0
	7	Animals	Lung	CBS 120.62	98.02%	99.82%	0
	8	Humans	Nasal swab	CBS 119368	98.41%	100.00%	0
	9	Humans	Nasal swab	CBS 119368	98.41%	100.00%	0
*A. niger*	10	Environment	Superficial swab	ITEM 11448	97.91%	100.00%	0
	11	Environment	Air	ITEM 10206	98.23%	100.00%	0
	12	Feed	Feed	CNRMA13.330	98.23%	99.64%	0

**Table 5 jof-08-01213-t005:** Results of antifungal susceptibility test.

Antimycotic Drug	Number of Total Resistant Strains (%)	Number And Type of Resistant Strains	MIC Value
Anidulafungin	6 (50%)	3 *A. fumigatus* 3 *A. flavus*	8 µg/mL 8 µg/mL
Micafungin	6 (50%)	3 *A. fumigatus* 3 *A. flavus*	8 µg/mL 8 µg/mL
Caspofungin	6 (50%)	3 *A. fumigatus* 3 *A. flavus*	8 µg/mL 8 µg/mL
5-fluorocytosine	8 (67%)	2 *A. fumigatus* 4 *A. flavus* 2 *A. niger*	32 µg/mL 64 µg/mL 32 µg/mL
Posaconazole	1 (8%)	1 *A. niger*	4 µg/mL
Voriconazole	2 (17%)	2 *A. niger*	2 µg/mL
Itraconazole	3 (25%)	2 *A. fumigatus* 1 *A. flavus*	1 µg/mL 2 µg/mL
Fluconazole	12 (100%)	4 *A. fumigatus* 5 *A. flavus* 3 *A. niger*	128µg/mL 64 µg/mL 64 µg/mL
Amphotericin B	1 (8%)	1 *A. flavus*	4 µg/mL

## Data Availability

Not applicable.

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
