# Peer review of "Preliminary Investigation about Aspergillus spp. Spread in Umbrian Avian Farms"

_jof, 2022, doi:10.3390/jof8111213_

Round 1

Reviewer 1 Report

The topic of the study is interesting and important, nevertheless  the presentation of susceptibility tests results  and their interpretation are controversial. Authors should present  MIC values obtained for individual species and provide an interpretation criterion (e.g. AFST_EUCAST_BP_ECOFF_v3.0_18_01_22.pdf). Finally  percentages of resistant strains could  also be presented. Resistance of Aspergillus to fluconazole is rather innate. 

Author Response

Dear Reviewer 1,

thank you for your generous comments on the manuscript and your comments and suggestions. The revisions have been approved by all authors and the changes are marked using the “Track Changes” function as requested. The revised manuscript has been uploaded.

Point 1: Authors should present MIC values obtained for individual species and provide an interpretation criterion (e.g. AFST_EUCAST_BP_ECOFF_v3.0_18_01_22.pdf). Finally  percentages of resistant strains could  also be presented. Resistance of Aspergillus to fluconazole is rather innate. 

Response 1: The MIC values have been added in Table 5: the interpretation criterion has been provided and it could be also found in the “References” section.

Reviewer 2 Report

In this preliminary investigation, the authors acquire information about the spread of Aspergillus spp., the sensitivity to antifungals and the presence of mycotoxins in some farms. This topic is interesting, but it has to be noted that the Identification and classification of strains are mainly depend on morphology on one medium, considering the shortcomings of this method, the results are somewhat inaccurate.

I have the followings for the authors to consider.

1)     Figure 1 and Figure 2 should go to the supplementary materials. Figure 3 is missing in the ms.

2)     Table 4, except for the direct blast information (similarity), a phylogenetic tree based on the sequencing information should be constructed to further confirm the results.

3)     Table 5, detailed values of MIC should be presented in this table.

Author Response

Dear Reviewer 2,

thank you for your comments and suggestions about the manuscript. The revisions have been approved by all authors and the changes are marked using the “Track Changes” function as requested. The revised manuscript has been uploaded.

The authors responded to Reviewer comments and suggestions as follows.

Point 1: Figure 1 and Figure 2 should go to the supplementary materials. Figure 3 is missing in the ms.

Response 1: Figure 1 and Figure 2 have been moved to the “Supplementary materials” section. The other Figures have been renumbered.  

Point 2: Table 4, except for the direct blast information (similarity), a phylogenetic tree based on the sequencing information should be constructed to further confirm the results.

Response 2: The phylogenetic tree has been constructed and added to the manuscript (Figure 4).

Point 3: Table 5, detailed values of MIC should be presented in this table.

Response 3: The MIC values have been added in Table 5.

Reviewer 3 Report

In this manuscript, the authors described studies on finding the spread of fungi, specifically the Aspergillus genus, in avian farms in a specific region in Italy.  They used genetic analysis to identify the specific Aspergillus species and HPLC to detect specific mycotoxins.

In general, the conclusions drawn were well supported by the results presented and the manuscript is well written. The preliminary study has important implication for understanding the potential risk of spreading and transmission of fungal spores and air-borne mycotoxins to workers operating in poultry.  However, the authors need to consider the following comments to improve their manuscript.

Specific comments:

1.       On page 4 line 113, “It was used Czapek agar (CZA), ….”   Need to correct this sentence.

2.       On page 5 line 154, what is ISO?

3.       The authors may or may not know that Aspergillus is highly used in industry and laboratory for research purposes.  E.g., A. niger and A. oryzae are used as tools in synthetic biology.  A. oryzae is widely used in fermentation in the food industry.  Therefore, these species clearly harmless.  The authors need to discuss the rationale for using Aspergillus in their study and whether we should be concerned about using A. oryzae and A. niger freely in labs and in industry.

Author Response

Dear Reviewer 3,

thank you for your generous comments on the manuscript and your comments and suggestions. The revisions have been approved by all authors and the changes are marked using the “Track Changes” function as requested. The revised manuscript has been uploaded.

The authors responded to Reviewer comments and suggestions as follows.

Point 1: On page 4 line 113, “It was used Czapek agar (CZA), ….”   Need to correct this sentence.

Response 1: The sentence has been corrected.

Point 2: On page 5 line 154, what is ISO?

Response 2: ISO, the  acronym of International Organization for Standardization, is the most important organization worldwide for the definition of technical standards.

Point 3: The authors may or may not know that Aspergillus is highly used in industry and laboratory for research purposes.  E.g., A. niger and A. oryzae are used as tools in synthetic biology.  A. oryzae is widely used in fermentation in the food industry.  Therefore, these species clearly harmless.  The authors need to discuss the rationale for using Aspergillus in their study and whether we should be concerned about using A. oryzae and A. niger freely in labs and in industry. 

Response 3: The authors discussed this concept in the “Discussion” section. Moreover, the Reference 27 has been added.  

Round 2

Reviewer 1 Report

Sensititre YeastOne YO10 are antimicrobial susceptibility plates for testing Candida species. The authors used them for testing Aspergillus spp.(which is not a yeast), what required modification of the standard procedure. Therefore, it is necessary to describe the exact methodology and method of reading. In the case of spore-forming filamentous fungi, it is important how the inoculum is obtained (there should be spores, not mycelium), its size and how the test is read. According to CLSI methodology the activity of echinocandins does not determine the MIC value, but the MEC, which requires the evaluation of growth morphology.

Thank you for correcting table 5. However, in my opinion, it is still unreadable. Since the number of strains is small (12), please show the MIC value and the interpretation adopted (R or S) for each strain, e.g. list the antimycotics in columns and the individual strains in rows and in the last row you can show summarized numbers of R/S isolates.

5-fluorocytosine and fluconazole belong to narrow-spectrum antimycotics, active mainly against yeasts, and therefore their activity against Aspergillus usually has not been reported in research literature. And certainly there are no designed standard cut-off values that would allow interpretation of drug susceptibility ( classifying a strain as sensitive or resistant based on MIC). Therefore, it is necessary for the authors to explain exactly how they interpreted their results against each Aspergillus species.

Finally, susceptibility data are not included in discussion, except of one sentence, about high percentages of resistance to common antifungals. In current human medicine the main problem are infection due to Aspergillus fumigatus resistant to mould –active triazoles – voriconazole, itraconazole, posaconazole, or isavuconazole. The source of such strains is very often the agricultural environment, they were also isolated from poultry. Please refer to this phenomenon.

Drug susceptibility testing is for only 12 strains out of nearly 200 isolated. Please consider whether it would make sense not to present these data in the current publication.

Reviewer 2 Report

The authors have addressed all my concrens.